

# A technique for rapid source apportionment applied to ambient organic aerosol measurements from the Thermal desorption Aerosol Gas chromatograph (TAG)

Yaping Zhang[1], Brent J. Williams[1], Allen H. Goldstein[2], Kenneth S. Docherty[3]*,

Jose L. Jimenez[3]

[1]Department of Energy, Environmental, and Chemical Engineering, Washington University in St. Louis, St. Louis, Missouri, USA

[2]Department of Environmental Science, Policy, & Management, University of California, Berkeley, California, USA

[3]Cooperative Institute for Research in the Environmental Sciences (CIRES) and Dept. of Chemistry & Biochemistry,

University of Colorado at Boulder, Boulder, Colorado, USA

*currently at: Alion Science and Technology, US EPA Office of Research and Development, Research Triangle Park, North Carolina, USA

*Correspondence to*: Brent J. Williams (brentw@wustl.edu)

**Abstract.** We present a rapid method for apportioning the sources of atmospheric organic aerosol composition measured by

gas chromatography/mass spectrometry methods. Here, we specifically apply this new analysis method to data acquired on a thermal desorption aerosol gas chromatograph (TAG) system. Gas chromatograms are divided by retention time into evenly spaced bins, within which the mass spectra are summed. A previous chromatogram binning method was introduced for the purpose of chromatogram structure deconvolution (e.g., major compound classes) (Zhang et al., 2014). Here we extend the method development for the specific purpose of determining aerosol samples' sources. Chromatogram bins are arranged

into an input data matrix for positive matrix factorization (PMF), where the sample number is the row dimension, and the mass spectra-resolved eluting time intervals (bins) are the column dimension. Then two-dimensional PMF can effectively do three-dimensional factorization on the three-dimensional TAG mass spectra data. The retention time shift of the chromatogram is corrected by applying the median values of the different peaks' shifts. Bin width affects chemical resolution, but does not affect PMF retrieval of the sources' time variations for low-factor solutions. A bin width smaller

than the maximum retention shift among all samples requires retention time shift correction. A six-factor PMF comparison among aerosol mass spectrometry (AMS), TAG binning, and conventional TAG compound integration methods shows that the TAG binning method performs similarly to the integration method. However, the new binning method incorporates the entirety of the data set and requires significantly less pre-processing of the data than conventional single compound identification and integration.





# 1 Introduction

Atmospheric aerosols can impact human health (Dominici et al., 2006; Gauderman et al., 2015), atmospheric visibility (Sun et al., 2006; Junjun et al., 2014), the water cycle, and climate change (IPCC, 2013). Anthropogenic and natural sources emit primary pollutants, which undergo atmospheric chemical and physical transformation to produce secondary pollutants.

Organic aerosols account for 20-70% of total fine aerosols ($PM_1$) (Jimenez et al., 2009; Murphy et al., 2006). Their chemical composition can comprise thousands of organic compounds, whose sources and transformations are not fully understood due to their complexity and dynamic chemical properties and gas/particle partitioning (Hallquist et al., 2009; Goldstein and Galbally, 2007). Many efforts have been made to measure and apportion the major chemical components and source attributions of these organic aerosols (Zhang et al., 2014; Ng et al., 2011; Williams et al., 2014).

The Aerodyne aerosol mass spectrometer (AMS) is a widely used instrument for aerosol analysis due to its capability to quantitatively characterize the size-resolved bulk composition of $PM_1$ (Canagaratna et al., 2007). AMS reports the bulk (also size-resolved) composition of $PM_1$ in the form of ensemble mass spectra, which are generated from the linear superposition of the mass spectra of individual compounds. Positive Matrix Factorization (PMF), a multivariate factor analysis method (Paatero, 1997; Ulbrich et al., 2009), is applied to the ensemble mass spectra, and deconvolves the spectra into several

factors with approximately constant mass spectra and consistent temporal behavior. Each of these factors can represent hundreds to thousands of organic compounds from a source or atmospheric process. The use of this technique has been growing rapidly in the last ten years due to its broad applicability (Zhang et al., 2011). However, AMS inherently has limited chemical resolution because it reports ensemble mass spectra with high fragmentation, and some important aspects of the sources and processes affecting OA are difficult to resolve using only AMS data. To obtain higher chemical resolution,

another online technique, called thermal desorption gas chromatograph (TAG), was combined with mass spectrometry (GC/MS) to separate and measure individual compounds (Williams et al., 2006).

TAG is a fully automated, field deployable instrument that can provide molecular level separation of organic aerosols with one hour time resolution to help identify specific aerosol source signatures and atmospheric transformation processes, e.g. through PMF analysis of a suite of individual integrated compounds (Williams et al., 2006). In 2008, Goldstein et al.

developed a two-dimensional gas chromatograph combined with an in-situ TAG collection system, which can speciate more organic compounds in atmospheric aerosols than TAG alone (Goldstein et al., 2008). In 2013, Semi-Volatile (SV)-TAG was developed to extend TAG's capability to include quantitative characterization of semi-volatile organic compounds (Zhao et al., 2013), and in 2014, Isaacman et al. introduced a technique for online chemical derivatization on the SV-TAG system to improve quantification of oxygenated molecules (Isaacman et al., 2014). Recently, a combined TAG-AMS instrument

which can simultaneously measure the bulk and speciated composition of organic aerosols has been presented (Williams et al., 2014). The advantage of providing speciated timelines for organic chemicals is significant, however the time required for chromatographic peak identification, integration, and confirmation of integration quality for hundreds of compounds limits the wider application of using a chromatographic approach for aerosol analysis. Additionally, a significant fraction of





GC/MS data is typically present as an unresolved complex mixture (UCM) when analyzing ambient aerosol samples (Williams et al., 2006, 2010, 2014, 2016) and peak integration methods typically ignore the material that is not resolved. Therefore, rapid techniques for comprehensively analyzing the complete chromatographically-separated mass spectral data, including UCM signal, may broaden the application of the various TAG measurement methods.

One such method was recently introduced, in which each chromatogram is evenly divided into bins and PMF is performed on the covariance of signal vs.retention time to deconvolute the chromatogram into homologous compound series, individual compounds, and multiple UCM components (Zhang et al., 2014). In order to pursue source apportionment, we initially attempted to input these deconvolution chromatogram factors to a second PMF analysis. However, such an approach does not achieve source apportionment results. For example, the factor of homologous compound series contains the full range of

compounds (such as $C_{12}$-$C_{40}$ alkanes). The second PMF cannot effectively separate different sources which may contain just a specific portion of homologous compound series (for example, one source may contain $C_{12}$-$C_{30}$ alkanes whereas another source contains $C_{31}$-$C_{40}$ alkanes).

Here, we present an alternative method specifically designed for source apportionment of ambient organics measured by TAG, where PMF is performed on the covariance of species from different sources to deconvolve the study period into

major contributing sources or aerosol transformation processes. We investigate the data matrix for binning mass spectra, the retention time shift correction, bin resolution, and compare the results of the method against those from factor analysis of conventional TAG resolved compound integration method and AMS.

## 2 Methods

### 2.1 TAG instrument

Williams et al. describe the TAG instrument in detail (Williams et al., 2006) as applied during the Study of Organic Aerosol at Riverside (SOAR) 2005. Briefly, particles with diameters less than 1.5 µm are humidified and impacted onto a collection and thermal desorption (CTD) cell at 30 °C. The CTD cell is then heated to 310 °C, and the particles are thermally desorbed into a helium carrier gas that transports them into a GC oven at 45 °C, where they re-condense onto the head of the GC column. After sample injection, the GC oven slowly heats to 310 °C, and the compounds eluting through a 30 m low-polarity

column are then detected by a quadrupole mass spectrometer.  The TAG is fully automated, achieves hourly time resolution, and can cycle between ambient samples (particles + adsorbing semivolatile gases), filtered samples (adsorbing semivolatile gases), denuded samples (particles only), cell blanks (no collection), and syringe injected liquid calibration standards.

### 2.2 PMF

For PMF analysis, the input data matrix, X, with dimensions of n rows and m columns, is factorized into two matrices – the

time series matrix G (n×p) and the chemical profile matrix F (p×m), where p is the number of factors – and a third matrix, the residual matrix, E (n×m):





$$X = GF + E,\qquad(1)$$

The PMF model is fitted by weighted least squares, with the weights based on the estimated uncertainties of the individual input matrix data points ($\sigma_{ij}$ being the estimated uncertainty for data point $x_{ij}$). Thus mathematical formulation of PMF is to minimize the following function, Q:

5 $Q = \sum_{i=1}^{n}\sum_{j=1}^{m}\frac{e_{ij}^{2}}{\sigma_{ij}^{2}}$, subject to $g_{ik} \geq 0$ and $f_{kj} \geq 0$, (2)

where $e_{ij}$, $g_{ik}$, and $f_{kj}$ are elements of matrices $E$, $G$, and $F$; and $\sigma_{ij}$ is the estimated uncertainties of $x_{ij}$, which is an element of matrix $X$ (Paatero, 1997).

In this paper, the PMF2 algorithm, a PMF model solver, was used for solving equation (1). A custom software tool (PMF Evaluation Tool, PET, version 2.06 (Ulbrich et al., 2009)) in Igor Pro (version 6.3, WaveMetrics, Inc.) was used to evaluate 10 PMF outputs and related statistics.

**2.3 Error estimation for the PMF model**

The PMF model is fitted by weighted least squares, and the weights are based on the estimated uncertainties of the input matrix data points. Here we discuss the uncertainty for each ion peak of TAG data. Generally, the uncertainty of an ion signal can be estimated as the square root of the number of ions counted, based on Poisson statistics (Allan et al., 2003), 15 which is referred to as the Ion Counting (IC) error method. Alternatively, the uncertainty can also be expressed as:

$$Unc = \begin{cases} 2 \times MDL, & if\ signal\ \leq MDL \\ \sqrt{(signal \times error\ fraction)^2 + (MDL)^2}, & if\ signal\ > MDL \end{cases},\qquad(3)$$

where error fraction is reported 10% for TAG ambient air samples (Williams et al., 2006;Williams et al., 2010); MDL is method detection limit. A detailed description about how to retrieve TAG MDL from ambient measurement is included in the supporting information. PMF on TAG bins' mass spectra was not sensitive to the choice of either of the two error 20 methods above (Zhang et al., 2014). Choosing a method depends on the availability of input data for different error methods. For example, TAG data used in this paper were measured by Agilent quadrupole mass spectrometer (5973 QMS), which does not report ion counts, but only an adjusted relative abundance. Thus the MDL error method is used here.

**2.4 Data analysis**

The original chromatogram binning method for the purpose of chromatogram structure deconvolution utilizes the TAG 25 dataset from the Study of Organic Aerosol at Riverside (SOAR) 2005 (Zhang et al., 2014). Further development here of the chromatogram binning method for the purpose of source apportionment uses the same dataset. Detailed information regarding the SOAR field site and auxiliary measurements can be found in Williams et al. (2010) and Docherty et al. (2008, 2011). The TAG SOAR sampling sequence has been described in detail previously (Zhang et al., 2014). In summary, the TAG system sampled ambient (gas + particle) and filtered ambient (gas only) data, from which the subtraction yielded 30 particle-only data (with GC column bleed signal also being subtracted by this method). The PMF analysis on binned particle-



only data is called TAG-Bin PMF. TAG-Bin PMF indicates binning method for source apportionment unless stated otherwise. In the original TAG source apportionment paper using individual resolved compounds, 123 major compounds were identified and integrated using the Agilent ChemStation software. A detailed description of the compounds' integration and the PMF on the integrated compounds are given in Williams et al. (2010). Here, the PMF results from analyzing the

dataset of the 123 resolved compounds is called TAG-Integrated PMF. In the discussion of bin resolution, two-factor solutions are chosen for AMS and TAG-Bin (Integrated) PMF results – Hydrocarbon-like OA (HOA) and Oxygenated OA (OOA) for comparison. In the comparison of source factors, six-factor solutions are chosen for TAG-Bin and TAG-Integrated, and are both compared to AMS six-component solutions, as reported by Docherty et al. (2011). Table 1 shows the six AMS components. The comparison between AMS and TAG is based on the assumption that the composition

measured by AMS and TAG at $PM_1$ and $PM_{1.5}$, respectively, are overlapped significantly.

## 3 Results and discussion

### 3.1 PMF binning data matrix for source apportionment

For source apportionment, PMF works on the covariance of samples collected at different times. Therefore, the row dimension of binning data for the PMF matrix is only the sample number, and the column dimension is the bins' mass

spectra in retention time order. Figure 1 shows the PMF binning data matrix for source apportionment (this study) compared to the previous PMF binning matrix for chromatogram structure (Zhang et al., 2014). In the binning method for source apportionment, the row dimension is the sample number from 1 to n, and in the column dimension the first bin's mass spectra ranges from 1 to m, the second bin's mass spectra from m+1 to 2×m, where m is mass spectra m/z index.

### 3.2 Retention time shift correction

By the nature of the data format used for source apportionment, where the retention time is in the column dimension of the PMF input data matrix, the PMF solution produces a signal chromatogram for each factor, with a fixed signal vs. retention time for the whole study period. However, this is not strictly true for the actual chromatogram samples as recorded, because their retention time will shift from sample to sample due to different sample mass loadings, chemical composition, aerosol water content, and column condition. Figure 2a shows example compound retention time shifts for the 1st, middle and end of

the study focus period. An approximately constant profile, which is an assumption of the PMF model, is required for a successful factor analysis. Therefore, the retention time shift along the sample number dimension should be corrected before PMF analysis. In this study we approximately correct retention time shift by calculating the median values of the retention time shifts along the retention time dimension and correct the retention times accordingly. The Figure 2b shows the peak retention time after shift correction using the median value: the peaks display a closer overlap after correction. While this

correction greatly improves the retention time shift, it is not an exact correction. As discussed in further detail in section 3.3, multiple scan points will be summed (binned), and retention time shift corrections become less necessary when using larger



bin widths. Therefore, with the combination of median value correction and large bin width, the retention time shift issue will be addressed.

Figure 3 shows the evolution of the median retention time shift during the study time period. The 123 major compounds, used in a prior source apportionment study (Williams et al., 2010) and representing a wide range of the nonpolar and polar compounds, were used to calculate these median values of retention time shifts. A positive median value means the chromatographic peaks shift to the right of the first sample (the elution runs slower), whereas a negative value means peaks shift to the left of the first sample (an earlier, or faster, elution). Almost all the samples in Figure 3 shift to the left of the first sample. In general, the median values show a linear relationship with the sample number during the study time period. The full range of median shifts among all samples is 13 scan points (corresponding to 36.4 seconds) and this shift is likely due to a slow change in the condition of the column as the study progressed. However, daily retention time variability is also observed. Figure 2S (in supplemental info.) shows the median variability with respect to the linear fitting line in Figure 3, and the total ion signal of the TAG samples during the study time period. The median variability is highly anti-correlated (R = -0.81) with the total signal of the TAG samples – a metric for aerosol mass loading on the TAG system. The elution runs slower when TAG has less mass loading, whereas the elution runs faster when TAG has a higher mass loading. Mass overloading of GC system corresponds to the saturation of column stationary phase, and can change the peak shape and retention time (Zenkevich and Pavlovskii, 2015). This can be explained by lowered interaction between each molecule in the samples and the stationary column phase when more molecules are present (in a larger sample), allowing material to pass through the column slightly faster.

PMF results with and without retention time shift correction are compared in detail for different bin widths in section 3.3, bin resolution. For future users, the internal standards, the external standards, or the major compounds in samples, all of which work well in automated integration software due to high signal-to-background ratios, can be used to estimate retention time shifts. If desired, additional retention time shift precision can be achieved by including both the long-term median shift from column condition as well as the daily shift due to sample size. However, it will be shown later that high precision retention time shifts would only be required if operating this PMF method with very high bin resolution, but are not necessary in most analyses of interest.

### 3.3 Bin resolution

The effects of different bin widths, with and without retention time (r.t.) shift correction, are compared here. Figure 4 shows the Pearson correlation coefficient difference ΔR (= with r.t. shift correction -without r.t. shift correction) of time series for four pairs – TAG-Bin HOA vs. AMS HOA, TAG-Bin OOA vs. AMS OOA, TAG-Bin HOA vs. CO, and TAG-Bin OOA vs. Ox. Carbon monoxide (CO) has been shown to correlate highly with primary organic aerosol concentrations during the SOAR study and odd oxygen ($O_x = O_3 + NO_2$) has been shown to correlate with secondary organic aerosol concentrations (Docherty et al., 2011). The ΔR values are all zero at bin width 52 (meaning there are 52 scan points summed within each single bin) for the four pairs, which means that retention time correction and no correction produce the same correlations.





For the correlations of the four pairs, the $\Delta R$ shows little increase ( 0.0075 on average) as the bin width decreases from 52 to 13, whereas the $\Delta R$ begins to increase (0.03 on average) from the bin width 13 to 2, and dramatically increases (0.46 on average) from the bin width 2 to 1. The reason $\Delta R$ begins to increase at the bin width 13 is that the retention time shift among all samples is 13, which is shown in Figure 3. The TAG-Bin with the bin width larger than the total retention time

5    shift is not sensitive to the retention time shift correction, whereas TAG-Bin with bin widths smaller than the total shift is sensitive to the correction, and bin width 1 (where each scan point is retained) is extremely sensitive to the retention time shift correction. In this case, without prior retention time shift correction, the user would certainly not want to use every MS scan point in a PMF analysis, and would need to exceed bin widths of 13 scan points to minimize the impact of retention time shifts on PMF results.

Figure 3S (in supplemental info.) shows the correlation coefficient R of the four pairs with retention time shift correction (all analyses below are performed using r.t. shift correction). R increases only slightly (0.04 on average) as the bin width decreases from 52 to 1. The smaller the bin width, with correspondingly higher chemical resolution, does not increase R very much for this simple two-factor PMF solution (that requires limited chemical resolution). The bin's mass spectrum with

different bin widths is an ensemble mass spectrum derived from the linear superposition of the different mass spectra of individual compounds. PMF, a multi-linear model, can deconvolve the ensemble mass spectra, such as AMS mass spectra and TAG bins' mass spectra, into groups of mass spectra, which provide chemical information on the sources and atmospheric aging processes. Thus, ideally, different bin widths, which affect chemical resolution, will not affect PMF performed to retrieve the factors' time series. The slight increase (0.04) of R for bin widths from 52 to 1 may be because

better PMF error estimation, and better PMF fits for small peaks will be obtained when a small bin width is used. This explanation is supported by the fact that R increases more for TAG-Bin OOA (0.06 on average) than for TAG-Bin HOA (0.015 on average) as bin width decreases from 52 to 1. The compounds in the TAG-Bin OOA group have overall lower signal peaks than the compounds in the HOA group do, and are better fit by small bin widths.

Theoretically, at least five scan points can define a peak; practically, more than 10 scan points are found in the compound peaks. Thus, for future users, bin width of more than five scan points are recommended because the smaller bin width requires significant computational power, and takes exponentially more time for PMF fitting. The retention time shift correction is strongly required when the total retention time shift among all samples is larger than the bin width.

### 3.4 Profiles of a six-factor solution to the TAG-Bin method

The six-factor PMF analysis with bin width equal to 5 scan points and retention time shift correction is applied to the TAG-Bin data matrix. According to the source apportionment of this TAG dataset by Williams et al. (2010) using integration method, the potential sources for TAG collected samples could be local vehicle, secondary organic aerosol, food cooking, biomass burning etc. Here we don't relate these factors to sources again. But the major compounds (and UCM) in each



factor's chromatogram are described here for later comparisons with TAG-Integrated and AMS results. As shown in Figure 5, the first factor (F1) contains highest contributions from: isopropyl myristate, alkanes (octadecane, nonadecane, pentadecane, eicosane, heptadecane, pristane), phenanthrene, diethyl phthalate, 1-penten-3-one,1-phenyl-, 2-propanol,1-chloro-,phosphate(3:1), etc. The resulting chromatogram of factor two (F2) is dominated by the compounds: 2-propanol,1-

5 chloro-,phosphate(3:1), dibutyl phthalate, 1,4-dioxaspiro[5,5]undecan-3-one, 2-pentadecanone,6,10,14-trimethyl, carboxylic acids (dodecanoic acid, undecanoic acid, decanoic acid), tris(3-chloropropyl)phosphate, 2(3H)-furanone,dihydro-5-decyl-, pelletierine, phthalimide, etc. Factor three (F3) contains major compounds: 2-propanol, 1-chloro-, phosphate (3:1), alkanes (nonadecane, heneicosane, eicosane, heptadecane), phthalic acid, phthalimide, phenanthrene, triacetin, dibutyl phthalate, decanoic acid, benzene,p-diacetyl-, etc. Factor four (F4) shows major resolved peaks mostly composed of oxygenated

molecules: 2-pentadecanone,6,10,14-trimethyl, phthalates (diisobutyl phthalate, dibutyl phthalate), carboxylic acids (nonanoic acid, decanoic acid, dodecanoic acid, hexadecanoic acid,methyl ester), phthalic acids (phthalic acid, methyl phthalic acid), triacetin, furanones (dihydro-5-heptyl-2(3H)-furanone , dihydro-5-octyl-2(3H)-furanone, dihydro-5-undecyl-2(3H)-furanone) etc. Besides the resolved peaks, F4 has a portion of UCM in the retention time range of 35-43 min, with the mass spectrum featuring m/z of 43, 55, 69, 81, 95, 109, etc. Factor five (F5)'s chromatogram has major compounds: 2-

pentadecanone,6,10,14-trimethyl, alkanes (nonadecane, octadecane, eicosane, heneicosane), phenanthrene, benzyl benzoate, 2-propanol,1-chloro-,phosphate(3:1), 1,4-dioxaspiro[5,5]undecan-3-one, phthalates (diisobutyl phthalate, dibutyl phthalate, diethyl phthalate), 2(3H)-furanone,dihydro-5-decyl-, nonanoic acid, pinonaldehyde, pelletierine, nonanal, benzoic acid, etc. Factor six (F6) shows major resolved peaks mostly composed of alkanes (C17-C29), isopropyl myristate, phenanthrene etc. In addition, F6 also contains a big portion of UCM in the retention time range of 41-51 min, with the mass spectrum dominated by m/z 43, 57, 69, 83, 97, 111 etc.

**3.5 Binning method for source apportionment compared to previously developed method for chromatogram deconvolution**

The chromatogram binning method has two applications for TAG data: chromatogram deconvolution described in detail in of Zhang et al. (2014), and source apportionment presented here. The PMF factor chromatograms and time series of both the

25 six- and 20-factor solutions are compared here. Figure 4S (in supplemental info.) displays the six-factor chromatograms and mass spectral profiles for the chromatogram deconvolution method. The twenty-factor chromatograms and mass spectral profiles for the chromatogram deconvolution method are presented in Zhang et al. (2014). In the six-factor solution for chromatogram deconvolution, two of the six factors are mainly resolved compounds (one is the alkane compound class, the other is mostly phthalic acid compound classes), and the others are dominantly composed of UCM. In the 20-factor solution

for chromatogram deconvolution, more compound classes are separated as single factors – alkanes, carboxylic acids, furanones, phthalates, cylcyclohexanes etc., as well as several individual compound factors. In the binning method for source apportionment, the six-factor solution was previously described in section 3.4. For the 20-factor solution, the compounds in each factor are marked in the Figure 5S (in supplemental info.). Compared to the previous binning method for chromatogram



deconvolution, this method for source apportionment tends to load many of the compounds into multiple factors since many of the compounds can be due to multiple sources. PMF factors resulting from the source apportionment method contain a greater diversity of compound types that correlate over sample time and represent a mixed chemical profile for specific source types or aerosol processes. The binning method for chromatogram deconvolution found major chromatogram

components and individual factors were dominated by major compound classes with similar mass spectral features (e.g., alkanes series, acid series, phthalate series, etc.).

The six-factor time series of the binning method for source apportionment (Table 2) and chromatogram deconvolution (Table 1S in supplemental info.) are compared to the six-factor time series of AMS PMF factors, which are considered as the source components. It is noted that due to the AMS instrument's quantitative ability we make comparisons to major

components of OA as determined by AMS PMF. However, the AMS measurement offers limited chemical separation compared to the TAG system and is not capable of determining many likely sources that contribute to atmospheric OA. While we utilize AMS PMF components here as an independent third method for comparison, it is likely possible that TAG measurements are capable of resolving additional sources or transformation processes compared to AMS. The maximum correlation coefficients (R) with AMS factors in each table (Table 2 and Table 1S) are summarized in Figure 6S. Most of

factors in the binning method for source apportionment display a better correlation with AMS factors than the factors in the chromatogram deconvolution method, indicating the binning method for source apportionment is superior for the purpose of source apportionment. Table 2S and 3S (in supplemental info.) show the correlation coefficients R of the AMS 6 components with the 20-factor solution of TAG-Bin for chromatogram deconvolution and source apportionment, respectively. While the binning method for chromatogram deconvolution displayed some high correlations with a few of the

AMS PMF components, this method used a 20-factor PMF solution to more completely separate chemical components, and resulting individual compound classes can have a high correlation with AMS components similar to how individual marker compounds can have a high correlation with AMS components. Here (Table 3S) it is observed that individual factors from the source apportionment method are more distinct and tend to correlate highest with a single AMS component, as opposed to correlating with multiple components as was observed in the chromatogram deconvolution method (Table 2S).

Additionally, there are some factors that correlate even with the minor AMS components (e.g., LOA-AC, LOA2, SV-OOA) when using the source apportionment method. To do an independent separation of major sources using only TAG data, this new binning method for source apportionment must be applied.

### 3.6 Source factor comparisons between AMS and the TAG binning method for source apportionment

Table 2 shows the Pearson correlation coefficient R of six-factor PMF time series between AMS and TAG-Bin source

apportionment method. Three pairs of TAG-Bin (assigned by factor number) and AMS factors display good correlations: R = 0.87 for F4 vs. MV-OOA, R = 0.80 for F6 vs. HOA, R = 0.63 for F5 vs. SV-OOA. Figure 6(a-c) and Figure 7S(a-c) (in supplementary info.) show the mass spectra comparison of those three pairs in two different ways. For all of the three pairs, TAG-Bin vs. AMS for m/z < 100 follow the line y = x, whereas for the m/z > 100, it is above the line y = x. The patterns of





mass spectra for m/z < 100 are similar between TAG-Bin and AMS. For m/z > 100, TAG-Bin is much higher than AMS (also suggested in Figure 7S in supplementary info.). The TAG system has been reported to have higher contributions from larger fragments when compared to AMS mass spectra, likely due to lower temperature of molecules during evaporation and fragmentation (Williams et al., 2014). Here, factor F4 (paired with MV-OOA) is mostly composed of oxygenated

compounds – carboxylic acids, phthalic acids, triacetin, furanones etc. Besides the resolved compounds, F4 also contains a portion of UCM with the similar mass spectrum to AMS MV-OOV (see Figure 5(d) and 7(d)). Factor F6 (paired with HOA) contains a suite of alkanes ($C_{17}$-$C_{29}$) as well as a portion of UCM with the similar mass spectrum to AMS HOA (see Figure 5(f) and 7(e)). Finally, factor F5 (paired with SV-OOA) contains a large number of semivolatile compounds: nonanoic acid, pinonaldehyde, pelletierine, nonanal, benzoic acid, etc.

**3.7 PMF profiles from six-factor solution of TAG-Integrated method**

The six-factor PMF is applied to the more traditional TAG-Integrated method. The factor number is assigned to each factor. Figure 8 shows the chemical profiles of the TAG-Integrated six-factor solution. Factor one (F1) is mostly composed of the hydrocarbons – alkanes, PAHs, cyclohexanes, etc. Factor two (F2) is dominated by larger alkanes. Factor three (F3) is featured by the oxygenated compounds - carboxylic acids, phthalic acids, furanones, etc. Factor four (F4) has the major

compounds – terpenes, xanthone, cyclopenta(d,e,f)phenanthrenone, 1,4-benzenediamine, N-(1,3-dimethylbutyl)-N'-phenyl-, etc. Factor five (F5) is dominated by the oxygenated compounds – phthalic acids, furanones, ketones, sulfur-chlorine-phosphorus-containing compounds, etc. Factor six (F6) has high loadings of the nitrogen-containing compounds, furanones, and ketones.

**3.8 Source factor comparisons between TAG binning and integration methods**

Just as Table 2 had listed the Pearson correlation coefficient R of six-factor PMF time series between AMS and TAG-Bin source apportionment method, Table 3 shows the R of six-factor time series between AMS and TAG-Integrated (conventional compound analysis). Factor numbers are assigned by each PMF analysis and factor numbers will not be reported in the same order for different PMF methods. Also, it is not expected that PMF results from different instruments (AMS vs. TAG) and different input data matrices from the same instrument (TAG-Bin vs. TAG-Integrated) will divide

covarying factors identically. The maximum R (in the column dimension of tables) with AMS factors in each table are displayed in Figure 9 for the purpose of comparing TAG-Bin source apportionment and TAG-Integrated results. For both comparisons in Table 2 and 3, four (LOA-AC, SV-OOA, MV-OOA and HOA) of the six maximum R in the column dimension are also the maximum R in the row dimension (AMS factor's maximum R with each TAG factor). Similar maximum correlation R for TAG-Bin and TAG-Integrated indicates that TAG-Bin source apportionment shows similar

performance to TAG-Integrated. The maximum R pairs with AMS MV-OOA are TAG-Bin F4 and TAG-Integrated F3, which share many of the same compounds – carboxylic acids, phthalic acids, and furanones. The maximum R pairs with AMS HOA are TAG-Bin F6 and TAG-Integrated F1. They also present many of the same compounds – alkanes and PAHs.



In addition, TAG-Bin F6 has better R with AMS HOA than TAG-Integrated F1, and TAG-Bin F4 has better R with MV-OOA than TAG-Integrated F3. The improved correlation (ΔR=0.09 on average) is because TAG-Bin F6 and F4 have a portion of UCM, with the mass spectra similar to the AMS HOA and MV-OOA, respectively. In addition, TAG-Bin and TAG-Integrated factors have good correlations (R > 0.6) with AMS MV-OOA, SV-OOA and HOA factors, suggesting that the TAG system as operated during the SOAR study was good at measuring components which are related to MV-OOA, SV-OOA and HOA. Furthermore, TAG-Bin and TAG-Integrated factors have lower correlations (R < 0.5) with AMS LOA-AC and LOA2 factors. LOA-AC and LOA2 factors accounts only for 7% of total AMS mass, and it is expected that PMF results from different instruments, such as TAG and AMS, would produce lower correlations for minor factors such as these. The maximum R pairs with AMS LOA-AC are TAG-Bin F1 and TAG-Integrated F6. The TAG-Integrated F6 has better R than TAG-Bin F1. Many nitrogen-containing compounds, which are highlighted in the TAG-Integrated method using the normalized abundance as the PMF input by each compound's maximum raw signal, are loaded into TAG-Integrated F6. However, those compounds that have low absolute signals in the raw chromatogram are buried in the chromatogram profiles of TAG-Bin method, which uses the raw signal as the PMF input. For the AMS LV-OOA factor, TAG-Bin and TAG-Integrated factors display mid-range correlations (0.5<R<0.6), as many compounds in the LV-OOA category likely either undergo thermal decomposition or do not transfer through the 30m TAG separation column.

Table 4 shows the direct comparison between TAG-Bin and TAG-Integrated, without any reference to AMS PMF results. Three pairs of TAG-Bin and TAG-Integrated have good correlations (R > 0.75). These three pairs are associated with MV-OOA, SV-OOA and HOA (AMS factors) as suggested in Table 2 and 3. The TAG system during the SOAR field study was good at measuring species in MV-OOA and SV-OOA and HOA categories, which have high relative abundance and mid to low polarity. There is only one pair with a low correlation (R = 0.07). Two potential reasons for this are presented here. Firstly, TAG-Bin used the whole chromatogram signal as PMF input; whereas TAG-Integrated only used the 123 resolved compound signals as input. Different signal (mass) input may affect how PMF resolves factors. Secondly, as mentioned above, small peaks present in chromatograms are amplified in the TAG-Integrated method, whereas the signal and variability present in these small peaks will be buried in the large, comprehensive signal that is utilized by the TAG-Bin method. This could also produce different factor solutions between TAG-Bin and TAG-Integrated.

Although the binning and integration methods have similar performance, the binning method requires limited data pre-processing and incorporates the entirety of the data set, allowing for a comprehensive and rapid method for utilizing chromatographically-separated mass spectral data in factor analyses for the purpose of organic aerosol source identification. When this binning method for source apportionment is applied to future ambient data sets, the user will need to determine the appropriate number of PMF factors to choose for a solution. Each data set will be different and ultimately the operator will need to use their own discretion by utilizing all information available. In general, too few factors will combine sources or transformation processes that share either chemical profile similarities or temporal similarities, and too many factors will





begin to cause "factor splitting" where what should have been a single component is divided into multiple components based on very minor differences. The original TAG compound integration PMF solution published by Williams et al. (2010) found that 9 factors best described the analyzed data set. Here we presented a six-factor solution and 20-factor solution to TAG PMF analysis using the binning method for source apportionment. This is an appropriate range to explore for urban,

suburban, to rural locations where you would expect at least 6 major OA source contributors or atmospheric transformation processes that would alter chemical profiles. Urban locations may contain 20 or more contributing sources, however with that many factors is likely that PMF would begin to cause factor splitting of major sources before separating some of the minor contributing sources. Previous AMS PMF analyses have used higher factor solutions to separate minor contributing sources, then manually recombined major factors that had been split by the high factor solution (e.g., Docherty et al. 2011).

This is an option for TAG PMF analyses as well, and given the enhanced chemical resolution of the TAG, additional contributing sources are expected to be identified.

## 4 Conclusions and implications

In the chromatogram binning method for source apportionment, the whole chromatogram was divided into evenly-spaced bins, within which mass spectra were summed to form a bin's mass spectrum. PMF was applied to separate the sources

according to their covariance. The row dimension of the PMF binning data matrix is the sample number, and the column dimension is mass spectra eluting time bins. The retention time shift with respect to the first sample was investigated in both the retention time and the sample number dimensions. The median value of the retention time shifts in each sample is used to correct the major shifts of the chromatographic peaks. The effects of different bin widths, with and without retention time shift correction, were compared. When the bin width was smaller than the retention time shift among all samples, the

retention time shift correction was required. Bin width, which affects chemical resolution, does not affect the PMF retrieval of the factors' time series for low-factor simple solutions. In multiple-source comparisons, the binning method had similar performance to the conventional compound integration method, but the binning method incorporates the entirety of the data set, can be fully automated, and requires limited data pre-processing prior to PMF analyses.

In the future, it will be of high interest to investigate if TAG provides additional factor/source resolution than AMS, including for polar species when using online derivatization to measure them. Two binning methods, for chromatogram structure (Zhang et al., 2014) or study-time structure (the source apportionment method presented here), have now been shown to operate well for the TAG GC-MS data, and the approach should be of interest for any measurement technique (mass spectrometry or spectroscopy) with an additional separation dimension(s) (volatility, hygroscopicity, electrical

mobility, etc.).



FUNDING

Brent J. Williams and Yaping Zhang were supported by NSF 1437933 and NSF 1554061; Allen Goldstein and SOAR TAG data collection were supported by California Air Resources Board (CARB) award 03-324.

5 Jose L. Jimenez and Kenneth Docherty were supported by EPA STAR 83587701-0 and DOE (BER/ASR) DE-SC0011105. This manuscript has not been reviewed by EPA and thus no endorsement should be inferred.

SUPPLEMENTAL MATERIAL

Supplemental data for this article can be accessed on the publisher's website.

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



**Table 1: AMS six components of PMF analysis by Docherty et al.**

| Factor | Full name | % contribution | OA |
|--------|-----------|----------------|-----|
| **cLV-OOA** | composite Low Volatility Oxygenated Organic Aerosol | 31.3 | |
| **LOA-AC** | Amine Containing Local OA | 4.4 | |
| **LOA-2** | Local-OA 2 | 2.6 | |
| **SV-OOA** | Semi-Volatile Oxygenated-OA | 14.4 | |
| **MV-OOA** | Medium-Volatile Oxygenated-OA | 30.2 | |
| **HOA** | Hydrocarbon-like OA | 13.8 | |

5 **Table 2: The correlation coefficient R of factors' time series of PMF six-factor solution between AMS and TAG-Bin. The TAG-Bin indicates the TAG binning method for source apportionment. The bin width of 5 scan points with retention time shift correction is used here.**

| | | **AMS 6 components** | | | | | |
|---|---|---|---|---|---|---|---|
| **Pearson R** | | cLV-OOA | LOA-AC | LOA2 | SV-OOA | MV-OOA | HOA |
| | *F1* | .08 | .38 | -.32 | .26 | -.03 | .31 |
| | *F2* | .45 | .01 | .11 | -.09 | .39 | .12 |
| **TAG-Bin 6 factors** | *F3* | -.15 | .26 | .11 | .05 | -.44 | -.32 |
| | *F4* | .55 | -.26 | -.26 | -.41 | .87 | -.12 |
| | *F5* | -.51 | -.20 | .20 | .63 | -.39 | -.05 |
| | *F6* | -.09 | -.01 | -.18 | -.15 | -.01 | .80 |
| **Maximum** | | .55 | .38 | .20 | .63 | .87 | .80 |



**Table 3: The correlation coefficient R of factors' time series of PMF six-factor solution between AMS and TAG-Integrated. The TAG-Integrated indicates TAG integration method.**

| Pearson R | | AMS 6 components | | | | | |
|---|---|---|---|---|---|---|---|
| | | cLV-OOA | LOA-AC | LOA2 | SV-OOA | MV-OOA | HOA |
| TAG-Integrated 6 factors | *F1* | .07 | .11 | -.17 | -.32 | -.01 | .71 |
| | *F2* | -.27 | -.03 | .08 | .19 | -.21 | .53 |
| | *F3* | .54 | -.22 | -.26 | -.31 | .78 | .06 |
| | *F4* | -.59 | -.27 | .05 | .63 | -.36 | -.02 |
| | *F5* | .46 | -.13 | -.03 | -.39 | .59 | -.58 |
| | *F6* | -.03 | .45 | .18 | .49 | -.44 | .21 |
| **Maximum** | | .54 | .45 | .18 | .63 | .78 | .71 |

**Table 4: The correlation coefficient R of factors' time series of PMF six-factor solution between TAG-Bin and TAG-Integrated. The TAG-Integrated indicates TAG integration method. The TAG-Bin indicates the TAG binning method for source apportionment. The bin width of 5 scan points with retention time shift correction is used here.**

| Pearson R | | TAG-Bin 6 factors | | | | | |
|---|---|---|---|---|---|---|---|
| | | *F1* | *F2* | *F3* | *F4* | *F5* | *F6* |
| TAG-Integrated 6 factors | *F1* | 0.2 | -0.05 | -0.21 | -0.02 | -0.18 | 0.88 |
| | *F2* | 0 | 0.23 | -0.51 | -0.17 | 0.35 | 0.57 |
| | *F3* | 0.01 | 0.36 | -0.51 | 0.76 | -0.4 | 0.2 |
| | *F4* | -0.08 | -0.43 | 0.07 | -0.5 | 0.84 | 0.05 |
| | *F5* | -0.02 | 0.06 | 0.06 | 0.68 | -0.46 | -0.44 |
| | *F6* | 0.33 | 0.48 | 0.01 | -0.44 | 0.17 | -0.12 |
| **Maximum** | | 0.33 | 0.48 | 0.07 | 0.68 | 0.84 | 0.88 |



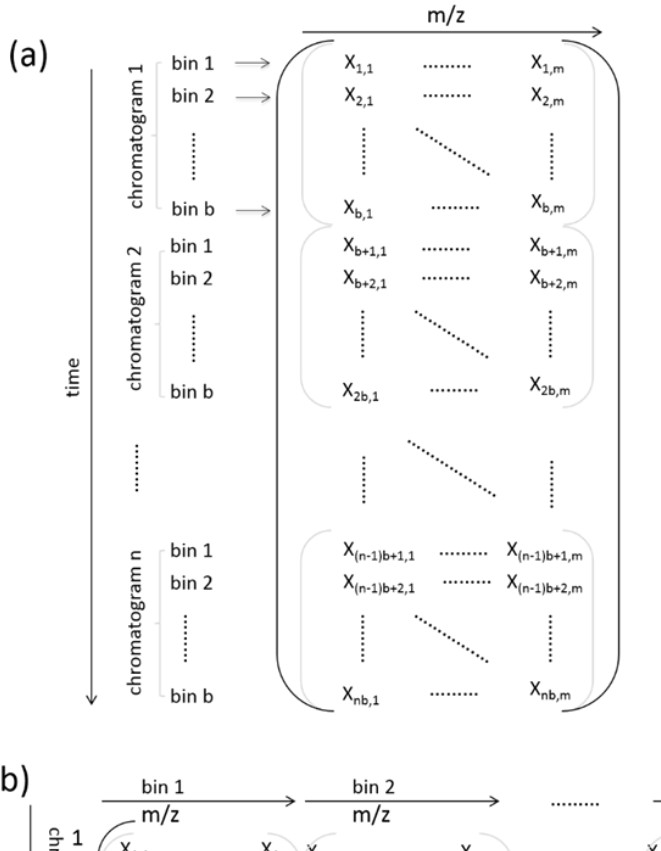

**Figure 1: PMF data matrices of (a) the binning method for deconvolution of chromatograms, and (b) source apportionment. The parameter n is the chromatogram number for each hour, b is the bin number, and m is the index of mass spectrum m/z.**





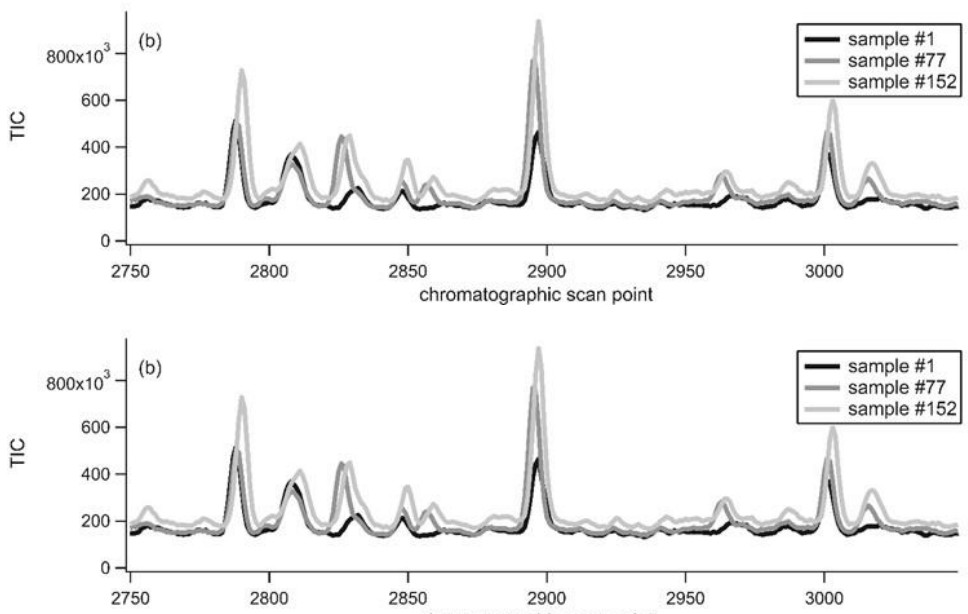

**Figure 2: Excerpts of chromatogram for the 1st, middle and end of the study focus period a) before retention time shift, and b) after retention time shift. The sample collection times for #1, 77, 152 are 7/29/2005 00:00, 8/3/2005 08:00, 8/8/2005 07:00, respectively.**




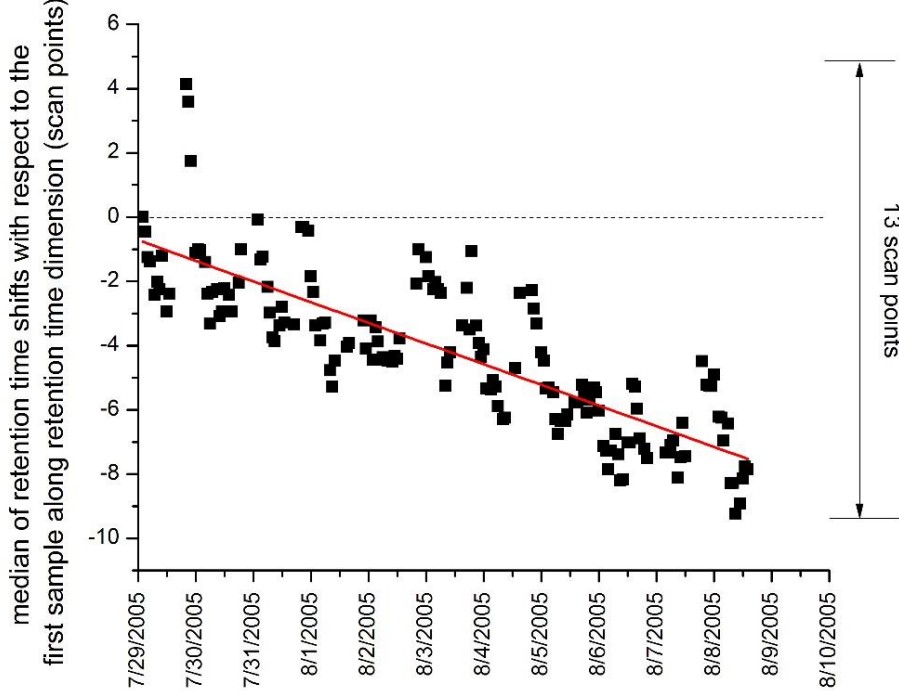

**Figure 3: Median values of retention time shifts with respect to the first sample. The positive median value means the chromatographic peaks shift to the right of the first sample (the elution runs slower), whereas the negative means peaks shift to the left (faster elution). The range of median shifts among all samples during the study period is 13 scan points.**





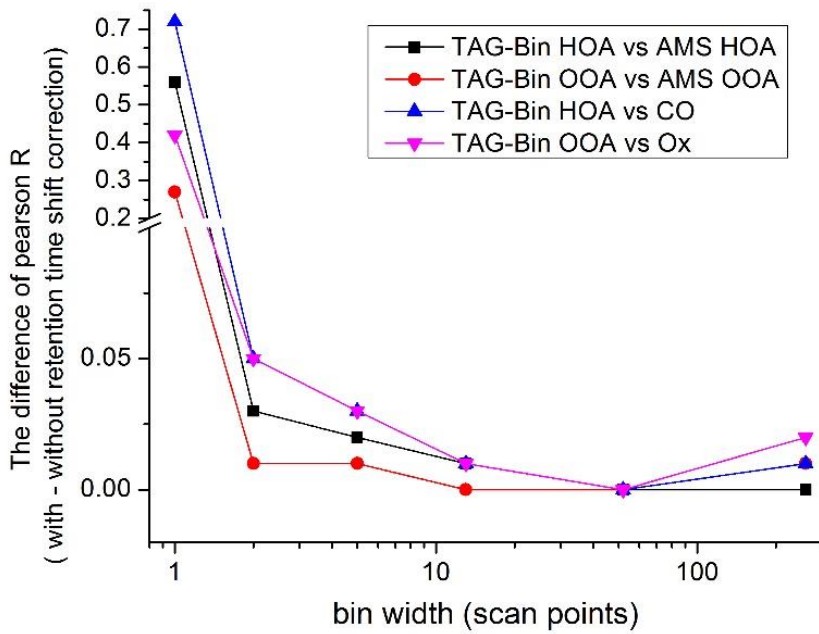

**Figure 4: The difference of Pearson R values obtained with retention time shift correction subtracted from values obtained with no retention time shift correction, of TAG-Bin HOA (OOA) time series compared to AMS HOA (OOA), and CO (Ox) time series.**



**(a)**

1 nonanal
levoglucosenone
.alpha.-campholenal
cyclopentasiloxane, decamethyl-
benzoic acid
p-methylacetophenone
dodecane
furan, 2-ethyl-5-methyl-
chrysanthenone
nopinone
nonanoic acid
pinonaldehyde
tridecane
naphthalene, 1-methyl-
phthalic acid
triacetin
decanoic acid
tetradecane
ethanone, 1-[4-(1-methylethenyl)phenyl]-
benzo[b]thiophene, 2-ethyl-5-methyl-
3-methylphthalic acid
4-methylphthalic acid
undecanoic acid

phthalimide
4-(1'-hydroxy-1'-methylethyl)acetophenone
pentadecane
dibenzofuran
dodecanoic acid
1-penten-3-one, 1-phenyl-
diethyl phthalate
1,4-dioxaspiro[5,5]undecan-3-one
benzophenone
heptadecane
pristane
ethylhexyl benzoate
hexyl cinnamic aldehyde
benzyl benzoate
octadecane
phenanthrene
2-propanol, 1-chloro-, phosphate (3:1)
isopropyl myristate
nonadecane
2(3H)-furanone,dihydro-5-decyl-
hexadecanoic acid, methyl ester
7,9-di-tert-butyl-1-oxaspiro(4,5)deca-6,9-diene-2,8-dione
eicosane

F1

**(b)**

nonanal
benzoic acid
furan, 2-ethyl-5-methyl-
nopinone
2(3H)-furanone,dihydro-5-butyl-
nonanoic acid
pinonaldehyde
tridecane
phthalic acid
decanoic acid
3-methylphthalic acid
benzene, p-diacetyl-
pelletierine
undecanoic acid
phthalimide
pentadecane

dodecanoic acid
1,4-dioxaspiro[5,5]undecan-3-one
heptadecane
benzyl benzoate
2-propanol, 1-chloro-, phosphate (3:1)
tris(3-chloropropyl)phosphate
2-Pentadecanone, 6,10,14-trimethyl-
diisobutyl phthalate
2(3H)-furanone,dihydro-5-decyl-
7,9-di-tert-butyl-1-oxaspiro(4,5)deca-6,9-diene-2,8-dione
dibutyl phthalate
eicosane
heneicosane
2(3H)-furanone, dihydro-5-dodecyl
docosane
tricosane
benzyl butyl phthalate

F2




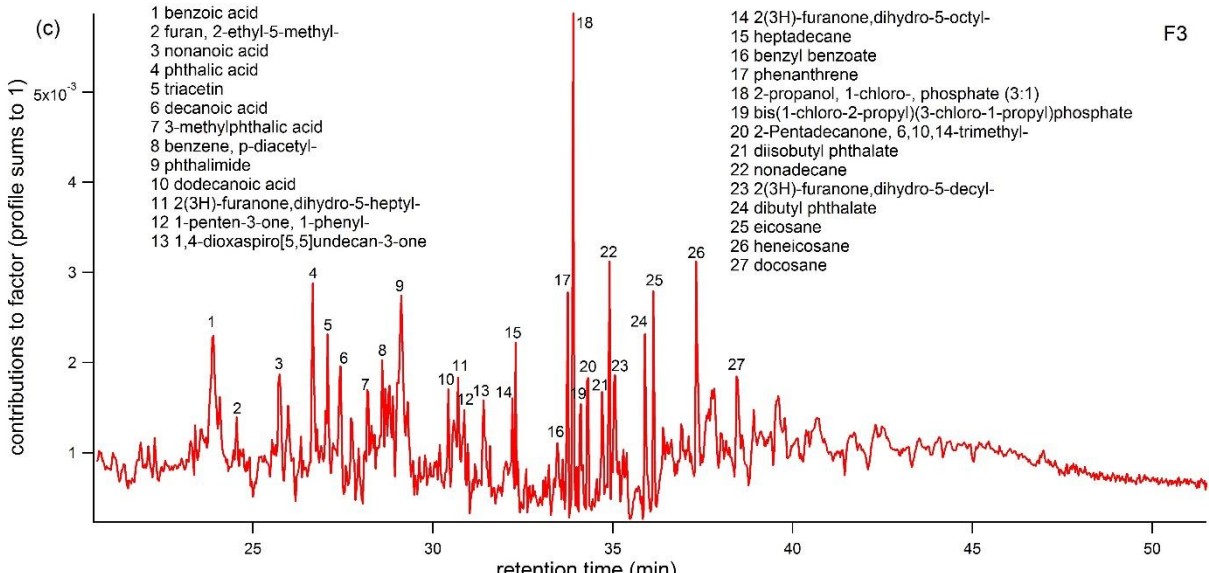

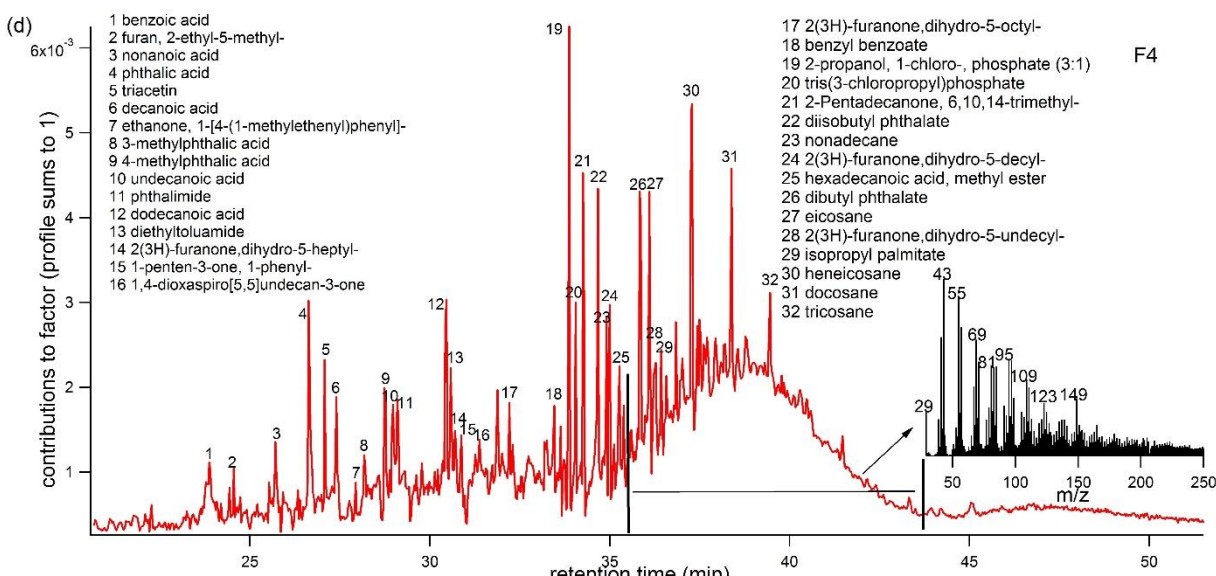





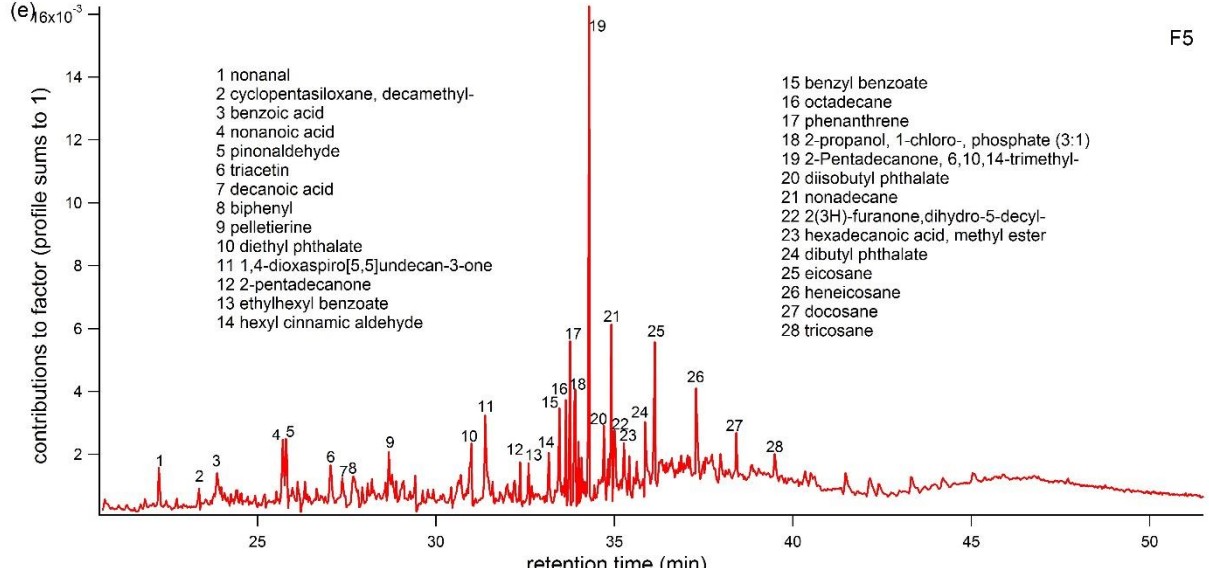

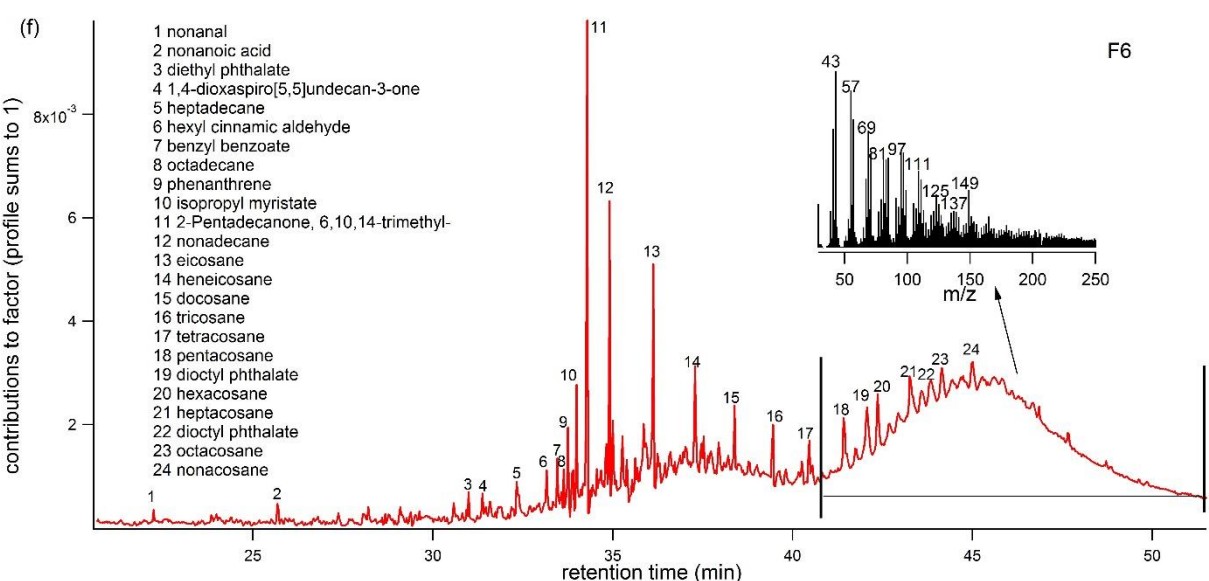

**Figure 5: The chromatogram profiles of TAG-Bin six-factor solution. The bin width of five scan points and retention time shift correction are used here. The mass spectra in panels d (factor 4) and f (factor 6) are the summed mass spectra of what are largely UCM components in the indicated retention time range.**







**Figure 6: The mass spectra comparison of the six pairs of TAG-Bin (assigned by factor number) vs. AMS – F4 vs. MV-OOA, F6 vs. HOA, F5 vs. SV-OOA, F4 vs. LV-OOA, F1 vs. LOA-AC, F5 vs. LOA2. Each pair shows the maximum Pearson R in Table 2.**
5 **The mass spectra (m/z 29 – 343 is shown in the color scale) is the normalized signal in log scale.**





**Figure 7: The mass spectra of AMS six components.**





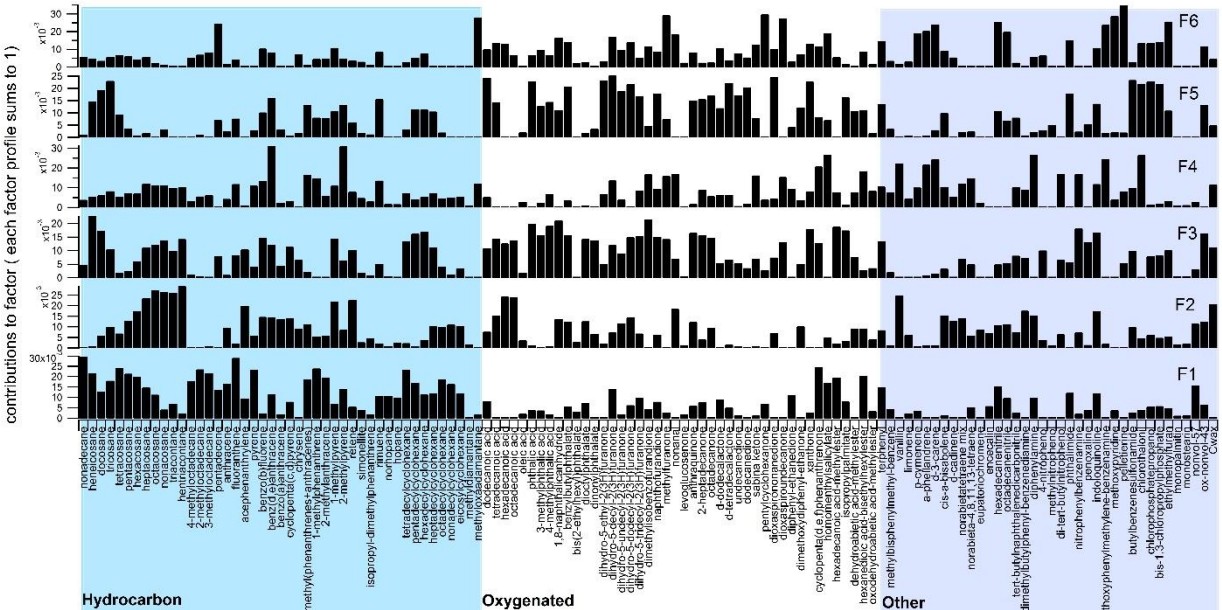

**Figure 8: Profiles of TAG-Integrated six-factor solution.**



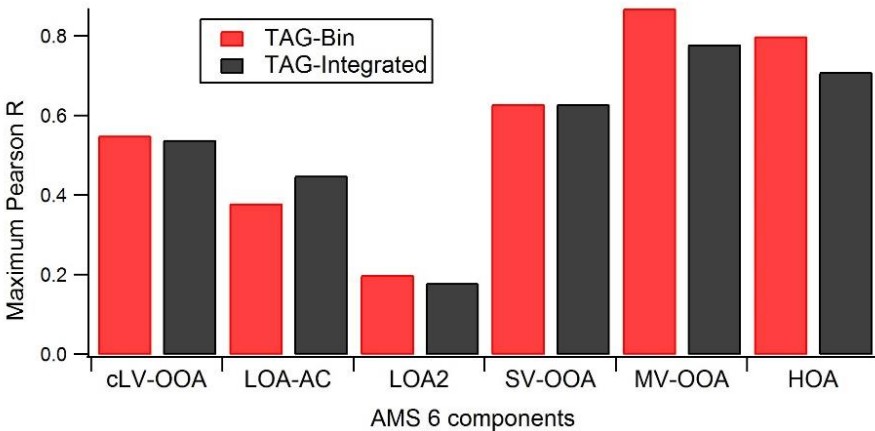

**Figure 9: Profiles of TAG-Integrated six-factor solution.**

