# Peer review of "A technique for rapid source apportionment applied to ambient organic aerosol measurements from the Thermal desorption Aerosol Gas chromatograph (TAG)"

_Atmospheric Measurement Techniques, 2016_

## Author Comment (AC1) · 28 Jul 2016

Figure 2(a) is missing and added here.

Figure 2. Excerpts of chromatogram for the 1st, middle and end of the study focus period a) before retention time shift, and b) after retention time shift. The sample collection times for #1, 77, 152 are 7/29/2005 00:00, 8/3/2005 08:00, 8/8/2005 07:00, respectively.
* * *
[Figure]

[Figure]

**Fig. 1.** Figure 2. Excerpts of chromatogram for the 1st, middle and end of the study focus period a) before retention time shift, and b) after retention time shift.

---

## Referee Comment (RC1) · Anonymous Referee #2 · 22 Aug 2016

This paper has introduced a very interesting approach of using GC/MS data for source apportionment. I agree with the authors that this approach should be of interest for other measurement techniques. However, I suggest more discussions about the following issues to help readers better understand this new approach, the PMF analysis of the TAG (GC/MS) data and sampling methods.

1) Correction for positive sampling artifacts. In this paper, the gas-phase inference on OA sampling was corrected for by subtracting organics collected downstream of a Teflon filter (vapors) from organics collected through a bypass line (vapors+particles).

[Figure]

Does the consistency in the PMF analysis of different data inputs suggest that this approach of correction for positive sampling artifacts is sufficient for source apportionment or to correct for gas-phase interference on OA sampling?

2) Organics measured by AMS and GC/MS. The thermal desorption temperature is 310°C for GC/MS and ∼600°C for AMS. Worton et al. (2014) has reported that GC/MS with the desorption temperature of 300°C only recover about 60% of organics collected in a traffic tunnel. Does this suggest that organics desorbed at 310°C can represent the variation of low-volatility organics?

D. R. Worton et al., Lubricating Oil Dominates Primary Organic Aerosol Emissions from Motor Vehicles. Environ. Sci. Technol. 48, 3698-3706 (2014).

3) Response factors of binned organics. The response factors of organic compounds are influenced by their functional groups, volatility and organic loadings. For quantification of individual species, the response factors of different compounds can be corrected for by internal standards. Have the variations in response factors of binned mass spectra been accounted for during the PMF analysis? Or these variations won't influence the results of the PMF analysis significantly?

4) On-line derivatization. This paper has shown that the PMF analysis of binned mass spectra in the retention time basis can resolve the SV-OOA and MV-OOA factors well. Do the authors think that the missing factor of cLV-OOA in the results from the PMF analysis of binned mass spectra can be identified by applying derivatization?

5) Comparison between the TAG-Binning method and the TAG-integration method. The strong correlations were not found for all PMF factors in results from PMF analysis of these two data formats. The binning method also provides limit chemical information compared to the TAG-integration method (Organic-tracer method). Do the authors think which one is a better method to identify major OA components for further emission controls?

---

## Referee Comment (RC2) · Anonymous Referee #1 · 29 Sep 2016

General comment:

The paper is sound, well written and very well structured. It presents a novel method for a fast source apportionment (SA) of ambient particulate matter. This methos is less time consuming than methods based on individual organic molecules, and has some advantages over fully "bulk" methods (such as PMF run on AMS readings). However, the advantages are not clearly stated (e.g. in the abstract and in the introduction). It would be very interesting to know about the authors' opinion on the advantages and disadvantages of this SA methodology. The paper presents an error estimation for the

method, as well as a coparison with other SA proceures. The methodology presented in the paper also addresses technical limitations of GC/MS in a satisfactory way.

Particular comments:

Abstract

Introduction At the end of the first paragraph, a sentence talks about the past efforts made to apportion the major chemical components as well as to attribute sources. There have been many attempts and approaches to do so, relying not on individual components, but rather on bulk properties. However, the references cited are quite recent. Since this is a major goal of the present paper, this subject deserves a more in-depth reference. page 2, lines 17-19: which are the aspects that the AMS cannot resolve and that the capacity to resolve single components of the TAG can? Please elaborate and cite relevant works.

Methods

Results and Discussion

page 8, lines 23-24: reword "... described in detail in of Zhang ..." to "... described in detail in Zhang ..."

Conclusions and Implications

---

## Author Comment (AC2) · 25 Oct 2016

*A revised document (with Track Changes enabled) has been provided to the editor.

Anonymous Referee #1

General comment: The paper is sound, well written and very well structured. It presents a novel method for a fast source apportionment (SA) of ambient particulate matter. This methos is less time consuming than methods based on individual organic molecules, and has some advantages over fully "bulk" methods (such as PMF run on AMS readings). However, the advantages are not clearly stated (e.g. in the abstract

and in the introduction). It would be very interesting to know about the authors' opinion on the advantages and disadvantages of this SA methodology. The paper presents an error estimation for the method, as well as a coparison with other SA proceures. The methodology presented in the paper also addresses technical limitations of GC/MS in a satisfactory way.

Author Response: The authors thank the reviewer for the comments and address further questions below. Regarding the advantages of TAG binning method over traditional integration method, we have previously stated in the abstract that it "the new binning method incorporates the entirety of the data set and requires significantly less pre-processing of the data than conventional single compound identification and integration." The following sentence regarding its advantage over AMS has been added to the abstract at page 1, line 29-31. "In addition, while a fraction of the most oxygenated aerosol does not elute through an underivatized TAG analysis, the TAG binning method does have the ability to achieve molecular level resolution on other bulk aerosol components commonly observed by the AMS."

Particular comments:

1) Abstract Introduction At the end of the first paragraph, a sentence talks about the past efforts made to apportion the major chemical components as well as to attribute sources. There have been many attempts and approaches to do so, relying not on individual components, but rather on bulk properties. However, the references cited are quite recent. Since this is a major goal of the present paper, this subject deserves a more in-depth reference.

Author Response: The following has been added to page 2, line 8-14.: "While inorganic ions, EC/OC, OC functional groups and trace metals from off-line filters analyses of atmospheric aerosol have been used for source apportionment (Chueinta et al. 2000; Ito et al. 2004; Lee et al. 1999; Ramadan et al. 2000; Ahlm et al. 2013), recently, high time resolution Aerodyne aerosol mass spectrometer (AMS) mass spectra and Aerosol

Chemical Speciation Monitor (ACSM) have been extensively used to determine the major components of ambient OA (Zhang et al., 2011; Ng et al. 2011). Online and offline measurements of molecular level marker molecules have also been used to apportion the major chemical components and source attributions of these organic aerosols (Schauer et al., 1996; Jaeckels et al. 2007; Zhang et al., 2014; Williams et al., 2010, 2014)."

2) page 2, lines 17-19: which are the aspects that the AMS cannot resolve and that the capacity to resolve single components of the TAG can? Please elaborate and cite relevant works.

Author Response: The following has been added to the page 2, line 26-33.: "The organic matter presenting similar mass spectra cannot be resolved by AMS. However, this type of material can be resolved by gas chromatography separation. For example, all alkanes show similar mass spectral patterns with dominant mass spectral peaks at m/z 43, 57, 71, 85, etc. AMS only can separate alkanes from other chemical classes with different functional groups, such as organic acids. AMS cannot separate different alkanes which are all grouped into one component called hydrocarbon-like organic aerosol (HOA) (Zhang et al., 2011), although various alkanes may come from different sources. However, TAG can resolve all alkanes through gas chromatography separation and preserve individual compound information to determine potential temporal variability differences (Williams et al., 2006, 2010)."

3) Methods Results and Discussion page 8, lines 23-24: reword "... described in detail in of Zhang ..." to "... described in detail in Zhang ..."

Author Response: changed in the revised manuscript

---

## Author Comment (AC3) · 25 Oct 2016

*A revised document (with Track Changes enabled) has been provided to the editor.

Anonymous Referee #2

General Comments This paper has introduced a very interesting approach of using GC/MS data for source apportionment. I agree with the authors that this approach should be of interest for other measurement techniques. However, I suggest more discussions about the following issues to help readers better understand this new approach, the PMF analysis of the TAG (GC/MS) data and sampling methods. Author

[Figure]

Response: The authors thank the reviewer for the comments and address further questions below.

1) Correction for positive sampling artifacts. In this paper, the gas-phase inference on OA sampling was corrected for by subtracting organics collected downstream of a Teflon filter (vapors) from organics collected through a bypass line (vapors+particles). Does the consistency in the PMF analysis of different data inputs suggest that this approach of correction for positive sampling artifacts is sufficient for source apportionment or to correct for gas-phase interference on OA sampling?

Author Response: Yes, this approach of correction is sufficient for both source apportionment and correcting gas-phase interference on OA sampling as has been applied in previous reports (Zhang et al., 2014; Williams et al., 2010), although subsequent field studies have deployed a diffusion denuder upstream of the collection cell as opposed to subtraction through this filter method. Table 2 shows the factor correlations of source apportionment between TAG-Bin and AMS. TAG instrument deployed here is good at measuring MV-OOA and HOA. Correlations for MV-OOA and HOA are 0.87 and 0.80, respectively. These correlations are higher than for the same PMF factors found from the TAG-Integrated method that targeted tracer species with minimal gas-phase interference, further suggesting this approach of correction is reasonable for source apportionment.

2) Organics measured by AMS and GC/MS. The thermal desorption temperature is 310âŮęC for GC/MS and âĹij600âŮęC for AMS. Worton et al. (2014) has reported that GC/MS with the desorption temperature of 300âŮęC only recover about 60% of organics collected in a traffic tunnel. Does this suggest that organics desorbed at 310âŮęC can represent the variation of low-volatility organics? D. R. Worton et al., Lubricating Oil Dominates Primary Organic Aerosol Emissions from Motor Vehicles. Environ. Sci. Technol. 48, 3698-3706 (2014).

Author Response: It is agreed that traditional GC/MS utilizing a traditional 30-m GC

column without derivatization only detects a portion of OA specifically missing highly oxygenated and low-volatility organics. The correlation (0.55) of low-volatility organics (cLV-OOA) between TAG-Bin and AMS in Table 2 supports the argument here. To solve this problem, the TAG research community has begun to apply online-derivatization in some cases (Isaacman et al., 2014), shorter GC column lengths to enhance recovery of oxygenated material (Martinez et al., 2016), and has now started to detect the thermal decomposition products that come from the heating of low-volatility and highly functionalized OA (Williams et al., 2016).

A statement has been added to the manuscript (Page 12, Line 6-10): "To address low detection of this analytically challenging OA fraction, subsequent TAG field deployments have applied a range of methods to increase detection through online-derivatization in some cases (Isaacman et al., 2014), shorter GC column lengths to enhance recovery of oxygenated material (Martinez et al., 2016), or in other cases the thermal decomposition products from heating of low-volatility and highly functionalized OA have been detected and analyzed (Williams et al., 2016)."

3) Response factors of binned organics. The response factors of organic compounds are influenced by their functional groups, volatility and organic loadings. For quantification of individual species, the response factors of different compounds can be corrected for by internal standards. Have the variations in response factors of binned mass spectra been accounted for during the PMF analysis? Or these variations won't influence the results of the PMF analysis significantly?

Author Response: The response of organic compounds are influenced by their functional group, volatility and organic loadings. Williams et al. (2010) injected standards of 13 compounds with different polarities and volatilities when samples were collected. The results show that the response shift among these 13 compounds from the start to the finish of the sampling period ranges from -61% to 13% with a mean -18%. Some of this drift is due to a uniform detector drift that affects all compounds equally (which can be easily corrected and therefore would not significantly impact PMF results since

there is equal impacts on compound time series variability), but the remaining drift can be due to impacts such as changes in column conditions over the course of a long study. Column conditions change more dramatically over longer study periods and impacts various compound classes differently. However, this standard injection method with the limited individual compounds, which was applied during this older study, is not sufficient for correcting the individual bins' signal drift in the binning method. Online internal standard injection is now possible (Isaacman et al., 2011) and a complex mixture of various polarity and volatility molecules would need to be analyzed as surrogate species to represent bin-response, allowing for a scale to interpolate and track bin-response drift.

While the bin signals haven't been corrected for signal drift here, the relatively high correlations of factors between TAG-Bin and AMS in Table 2 are consistent with the fact that GC/MS (TAG) deployed here is particularly performing well at measuring MV-OOA and HOA type aerosol. The source apportionment using uncorrected signal is still reliable in the qualitative perspective analyzed here. Correction of signal drift for future studies is expected to further improve correlations between TAG-Bin and AMS OA components.

We've added the following statement to Page 7, Line 10-16: "In addition to retention time shifts, detector response and GC column conditions can drift over the course of a study. Individual bin response factors have not been developed for this data set due to limited calibration standards applied during the SOAR study. Online internal standard injections are now possible (Isaacman et al., 2011) and a complex mixture of various polarity and volatility molecules would need to be analyzed as surrogate species to represent bin response, allowing for a scale to interpolate and track bin response drift corrections. Given the shorter study focus period analyzed here and the relatively high correlations observed below between several TAG-Bin components and AMS components, the bin response correction does not appear critical here, but should be included in future applications of this method."

And again in the conclusions: "Future applications of this method should continue to apply retention time shifts when necessary and should incorporate new relationships using regularly injected calibration standards to develop bin-specific response factors, especially when longer study periods susceptible to larger drifts are to be analyzed."

4) On-line derivatization. This paper has shown that the PMF analysis of binned mass spectra in the retention time basis can resolve the SV-OOA and MV-OOA factors well. Do the authors think that the missing factor of cLV-OOA in the results from the PMF analysis of binned mass spectra can be identified by applying derivatization?

Author Response: Yes, that is the case. This point has been previously addressed in our response to Comment #2 above.

5) Comparison between the TAG-Binning method and the TAG-integration method. The strong correlations were not found for all PMF factors in results from PMF analysis of these two data formats. The binning method also provides limit chemical information compared to the TAG-integration method (Organic-tracer method). Do the authors think which one is a better method to identify major OA components for further emission controls?

Author Response: Firstly, we don't expect strong correlations for all the PMF factors between TAG-Binning and TAG-integration methods. These two methods incorporate different inputs in the PMF analysis - the whole chromatogram signal, and 123 integrated individual compounds, respectively. However, there should be significant overlap where the same type of variability was captured. In addition to the binning method requiring much less pre-processing compared to the traditional peak integration method, it also includes all of the GC/MS signal and does not risk missing an important compound or series of compounds as could easily occur for the traditional single-compound method. In terms of chemical resolution, the binning method can get down to the molecular level with appropriate retention time shifting. In this case, the operator could then go back and identify important compounds within each of the factors, after PMF analysis

has decided they are important species for a resulting component. We feel that this method will evolve and has the potential to replace the traditional method in the future. It is worth noting that higher bin resolution will also require more computer processing power, which should also become more manageable in the future.

The following statement has been added to Page 12, Line 26-30: "By incorporating all of the GC/MS signal, the binning method does not risk missing an important compound or series of compounds as could easily occur for the traditional single-compound method since the input compounds are chosen by the operator. In terms of chemical resolution, the binning method can get down to the molecular level with appropriate retention time shifting. In this case, the operator can then go back and identify important compounds within each of the factors, after PMF analysis has decided they are defining species for a resulting component."

---

## Author Comment (AC4) · 25 Oct 2016

**A technique for rapid source apportionment applied to ambient organic aerosol measurements from the Thermal desorption Aerosol Gas chromatograph (TAG)**

Yaping Zhang[1], Brent J. Williams[1], Allen H. Goldstein[2], Kenneth S. Docherty[3]*,
Jose L. Jimenez[3]

[1]Department of Energy, Environmental, and Chemical Engineering, Washington University in St. Louis, St. Louis, Missouri, USA

[2]Department of Environmental Science, Policy, & Management, University of California, Berkeley, California, USA

[3]Cooperative Institute for Research in the Environmental Sciences (CIRES) and Dept. of Chemistry & Biochemistry, University of Colorado at Boulder, Boulder, Colorado, USA

*currently at: Alion Science and Technology, US EPA Office of Research and Development, Research Triangle Park, North Carolina, USA

*Correspondence to*: Brent J. Williams (brentw@wustl.edu)

**Abstract.** We present a rapid method for apportioning the sources of atmospheric organic aerosol composition measured by gas chromatography/mass spectrometry methods. Here, we specifically apply this new analysis method to data acquired on a thermal desorption aerosol gas chromatograph (TAG) system. Gas chromatograms are divided by retention time into evenly spaced bins, within which the mass spectra are summed. A previous chromatogram binning method was introduced for the purpose of chromatogram structure deconvolution (e.g., major compound classes) (Zhang et al., 2014). Here we extend the method development for the specific purpose of determining aerosol samples' sources. Chromatogram bins are arranged into an input data matrix for positive matrix factorization (PMF), where the sample number is the row dimension, and the mass spectra-resolved eluting time intervals (bins) are the column dimension. Then two-dimensional PMF can effectively do three-dimensional factorization on the three-dimensional TAG mass spectra data. The retention time shift of the chromatogram is corrected by applying the median values of the different peaks' shifts. Bin width affects chemical resolution, but does not affect PMF retrieval of the sources' time variations for low-factor solutions. A bin width smaller than the maximum retention shift among all samples requires retention time shift correction. A six-factor PMF comparison among aerosol mass spectrometry (AMS), TAG binning, and conventional TAG compound integration methods shows that the TAG binning method performs similarly to the integration method. However, the new binning method incorporates the entirety of the data set and requires significantly less pre-processing of the data than conventional single compound identification and integration. In addition, while a fraction of the most oxygenated aerosol does not elute through an underivatized TAG analysis, the TAG binning method does have the ability to achieve molecular level resolution on other bulk aerosol components commonly observed by the AMS.

**1 Introduction**

Atmospheric aerosols can impact human health (Dominici et al., 2006; Gauderman et al., 2015), atmospheric visibility (Sun et al., 2006; Junjun et al., 2014), the water cycle, and climate change (IPCC, 2013). Anthropogenic and natural sources emit primary pollutants, which undergo atmospheric chemical and physical transformation to produce secondary pollutants.

Organic aerosols account for 20-70% of total fine aerosols ($PM_1$) (Jimenez et al., 2009; Murphy et al., 2006). Their chemical composition can comprise thousands of organic compounds, whose sources and transformations are not fully understood due to their complexity and dynamic chemical properties and gas/particle partitioning (Hallquist et al., 2009; Goldstein and Galbally, 2007).  While inorganic ions,

EC/OC, OC functional groups and trace metals from off-line filters analyses of atmospheric aerosol have been used for source apportionment (Chueinta et al. 2000; Ito et al. 2004; Lee et al. 1999; Ramadan et al. 2000; Ahlm et al. 2013), recently, high time resolution Aerodyne aerosol mass spectrometer (AMS) mass spectra and Aerosol Chemical Speciation Monitor (ACSM) have been extensively used to determine the major components of ambient OA (Zhang et al., 2011; Ng et al. 2011). Online and offline measurements of molecular level marker molecules have also been used to apportion the major chemical components and source attributions of these organic aerosols (Schauer et al., 1996; Jaeckels et al. 2007; Zhang et al., 2014; Williams et al., 2010, 2014).

The Aerodyne aerosol mass spectrometer (AMS) is a widely used instrument for aerosol analysis due to its capability to quantitatively characterize the size-resolved bulk composition of $PM_1$ (Canagaratna et al., 2007). AMS reports the bulk (also size-resolved) composition of $PM_1$ in the form of ensemble mass spectra, which are generated from the linear superposition of the mass spectra of individual compounds. Positive Matrix Factorization (PMF), a multivariate factor analysis method (Paatero, 1997; Ulbrich et al., 2009), is applied to the ensemble mass spectra, and deconvolves the spectra into several factors with approximately constant mass spectra and consistent temporal behavior. Each of these factors can represent hundreds to thousands of organic compounds from a source or atmospheric process. The use of this technique has been growing rapidly in the last ten years due to its broad applicability (Zhang et al., 2011). However, AMS inherently has limited chemical resolution because it reports ensemble mass spectra with high fragmentation, and some important aspects of the sources and processes affecting OA are difficult to resolve using only AMS data. To obtain higher chemical resolution, another online technique, called thermal desorption gas chromatograph (TAG), was combined with mass spectrometry (GC/MS) to separate and measure individual compounds (Williams et al., 2006). The organic matter presenting similar mass spectra cannot be resolved by AMS. However, this type of material can be resolved by gas chromatography separation. For example, all alkanes show similar mass spectral patterns with dominant mass spectral peaks at m/z 43, 57, 71, 85, etc. AMS only can separate alkanes from other chemical classes with different functional groups, such as organic acids. AMS cannot separate different alkanes which are all grouped into one component called hydrocarbon-like organic aerosol (HOA) (Zhang et al., 2011), although various alkanes may come from different sources. However, TAG can resolve all alkanes through gas chromatography separation and preserve individual compound information to determine potential temporal variability differences (
[revised manuscript text omitted]

In addition to retention time shifts, detector response and GC column conditions can drift over the course of a study. Individual bin response factors have not been developed for this data set due to limited calibration standards applied during the SOAR study. Online internal standard injections are now possible (Isaacman et al., 2011) and a complex mixture of various polarity and volatility molecules would need to be analyzed as surrogate species to represent bin response, allowing for a scale to interpolate and track bin response drift corrections. Given the shorter study focus period analyzed here and the relatively high correlations observed below between several TAG-Bin components and AMS components, the bin response correction does not appear critical here, but should be included in future applications of this method.

[revised manuscript text omitted]
. To address low detection of this analytically challenging OA fraction, subsequent TAG field deployments have applied a range of methods to increase detection through online-derivatization in some cases (Isaacman et al., 2014), shorter GC column lengths to enhance recovery of oxygenated material (Martinez et al., 2016), or in other cases the thermal decomposition products from heating of low-volatility and highly functionalized OA have been detected and analyzed (Williams et al., 2016).

Table 4 shows the direct comparison between TAG-Bin and TAG-Integrated, without any reference to AMS PMF results. Three pairs of TAG-Bin and TAG-Integrated have good correlations (R > 0.75). These three pairs are associated with MV-OOA, SV-OOA and HOA (AMS factors) as suggested in Table 2 and 3. The TAG system during the SOAR field study was good at measuring species in MV-OOA and SV-OOA and HOA categories, which have high relative abundance and mid to low polarity. There is only one pair with a low correlation (R = 0.07). Two potential reasons for this are presented here. Firstly, TAG-Bin used the whole chromatogram signal as PMF input; whereas TAG-Integrated only used the 123 resolved compound signals as input. Different signal (mass) input may affect how PMF resolves factors. Secondly, as mentioned above, small peaks present in chromatograms are amplified in the TAG-Integrated method, whereas the signal and variability present in these small peaks will be buried in the large, comprehensive signal that is utilized by the TAG-Bin method. This could also produce different factor solutions between TAG-Bin and TAG-Integrated.

Although the binning and integration methods have similar performance, the binning method requires limited data pre-processing and incorporates the entirety of the data set, allowing for a comprehensive and rapid method for utilizing chromatographically-separated mass spectral data in factor analyses for the purpose of organic aerosol source identification. By incorporating all of the GC/MS signal, the binning method does not risk missing an important compound or series of compounds as could easily occur for the traditional single-compound method since the input compounds are chosen by the operator. In terms of chemical resolution, the binning method can get down to the molecular level with appropriate retention time shifting. In this case, the operator can then go back and identify important compounds within each of the factors, after PMF analysis has decided they are defining species for a resulting component.

[revised manuscript text omitted]

Ahlm, L., Shakya, K. M., Russell, L. M., Schroder, J. C., Wong, J. P. S., Sjostedt, S. J., Hayden, K. L., Liggio, J., Wentzell, J. J. B., Wiebe, H. A., Mihele, C., Leaitch, W. R., Macdonald, A. M. (2013). Temperature-dependent accumulation mode particle and cloud nuclei concentrations from biogenic sources during WACS 2010. Atmos. Chem. Phys. 13:3393-3407.

Canagaratna, M. R., Jayne, J. T., Jimenez, J. L., Allan, J. D., Alfarra, M. R., Zhang, Q., Onasch, T. B., Drewnick, F., Coe, 25 H., Middlebrook, A., Delia, A., Williams, L. R., Trimborn, A. M., Northway, M. J., DeCarlo, P. F., Kolb, C. E., Davidovits, P., and Worsnop, D. R.: Chemical and microphysical characterization of ambient aerosols with the aerodyne aerosol mass spectrometer, Mass Spectrometry Reviews, 26, 185-222, 10.1002/mas.20115, 2007.

Chueinta, W., Hopke, P. K., Paatero, P. (2000). Investigation of sources of atmospheric aerosol at urban and suburban residential areas in Thailand by positive matrix factorization. Atmos. Environ. 34:3319-3329.

Docherty, K. S., Aiken, A. C., Huffman, J. A., Ulbrich, I. M., DeCarlo, P. F., Sueper, D., Worsnop, D. R., Snyder, D. C., Peltier, R. E., Weber, R. J., Grover, B. D., Eatough, D. J., Williams, B. J., Goldstein, A. H., Ziemann, P. J., and Jimenez, J. L.: The 2005 Study of Organic Aerosols at Riverside (SOAR-1): instrumental intercomparisons and fine particle composition, Atmos. Chem. Phys., 11, 12387-12420, 10.5194/acp-11-12387-2011, 2011.

Dominici, F., Peng, R. D., Bell, M. L., Pham, L., McDermott, A., Zeger, S. L., and Samet, J. M.: Fine particulate air pollution and hospital admission for cardiovascular and respiratory diseases, JAMA-J. Am. Med. Assoc., 295, 1127-1134, 10.1001/jama.295.10.1127, 2006.

Gauderman, W. J., Urman, R., Avol, E., Berhane, K., McConnell, R., Rappaport, E., Chang, R., Lurmann, F., and Gilliland, F.: Association of Improved Air Quality with Lung Development in Children, New England Journal of Medicine, 372, 905-913, doi:10.1056/NEJMoa1414123, 2015.

Goldstein, A. H., and Galbally, I. E.: Known and unexplored organic constituents in the earth's atmosphere, Environ. Sci. Technol., 41, 1514-1521, 10.1021/es072476p, 2007.

Goldstein, A. H., Worton, D. R., Williams, B. J., Hering, S. V., Kreisberg, N. M., Panic, O., and Gorecki, T.: Thermal desorption comprehensive two-dimensional gas chromatography for in-situ measurements of organic aerosols, J. Chromatogr. A, 1186, 340-347, 10.1016/j.chroma.2007.09.094, 2008.

Hallquist, M., Wenger, J. C., Baltensperger, U., Rudich, Y., Simpson, D., Claeys, M., Dommen, J., Donahue, N. M., George, C., Goldstein, A. H., Hamilton, J. F., Herrmann, H., Hoffmann, T., Iinuma, Y., Jang, M., Jenkin, M. E., Jimenez, J. L., Kiendler-Scharr, A., Maenhaut, W., McFiggans, G., Mentel, T. F., Monod, A., Prevot, A. S. H., Seinfeld, J. H., Surratt, J. D., Szmigielski, R., and Wildt, J.: The formation, properties and impact of secondary organic aerosol: current and emerging issues, Atmos. Chem. Phys., 9, 5155-5236, 2009.

Ito, K., Xue, N., Thurston, G. (2004). Spatial variation of PM2.5 chemical species and source-apportioned mass concentrations in New York City. Atmospheric Environment 38:5269-5282.

Isaacman, G., Kreisberg, N. M., Worton, D. R., Hering, S. V., Goldstein, A. H. (2011). A versatile and reproducible automatic injection system for liquid standard introduction: application to in-situ calibration. Atmospheric Measurement Techniques 4:1937-1942.

Isaacman, G., Kreisberg, N. M., Yee, L. D., Worton, D. R., Chan, A. W. H., Moss, J. A., Hering, S. V., and Goldstein, A. H.: Online derivatization for hourly measurements of gas- and particle-phase semi-volatile oxygenated organic compounds by thermal desorption aerosol gas chromatography (SV-TAG), Atmos. Meas. Tech., 7, 4417-4429, 10.5194/amt-7-4417-2014, 2014.

Jimenez, J. L., Canagaratna, M. R., Donahue, N. M., Prevot, A. S. H., Zhang, Q., Kroll, J. H., DeCarlo, P. F., Allan, J. D., Coe, H., Ng, N. L., Aiken, A. C., Docherty, K. S., Ulbrich, I. M., Grieshop, A. P., Robinson, A. L., Duplissy, J., Smith, J. D., Wilson, K. R., Lanz, V. A., Hueglin, C., Sun, Y. L., Tian, J., Laaksonen, A., Raatikainen, T., Rautiainen, J., Vaattovaara, P., Ehn, M., Kulmala, M., Tomlinson, J. M., Collins, D. R., Cubison, M. J., Dunlea, E. J., Huffman, J. A., Onasch, T. B., Alfarra, M. R., Williams, P. I., Bower, K., Kondo, Y., Schneider, J., Drewnick, F., Borrmann, S., Weimer, S., Demerjian, K., Salcedo, D., Cottrell, L., Griffin, R., Takami, A., Miyoshi, T., Hatakeyama, S., Shimono, A., Sun, J. Y., Zhang, Y. M., Dzepina, K., Kimmel, J. R., Sueper, D., Jayne, J. T., Herndon, S. C., Trimborn, A. M., Williams, L. R., Wood, E. C., Middlebrook, A. M., Kolb, C. E., Baltensperger, U., and Worsnop, D. R.: Evolution of Organic Aerosols in the Atmosphere, Science, 326, 1525-1529, 10.1126/science.1180353, 2009.

Junjun, D., Zhenyu, X., Bingliang, Z., and Ke, D.: Comparative study on long-term visibility trend and its affecting factors on both sides of the Taiwan Strait, Atmos. Res., 143, 266-278, 10.1016/j.atmosres.2014.02.018, 2014.

Jaeckels, J. M., Bae, M. S., Schauer, J. J. (2007). Positive matrix factorization (PMF) analysis of molecular marker measurements to quantify the sources of organic aerosols. Environ. Sci. Technol. 41:5763-5769.

Lee, E., Chan, C. K., Paatero, P. (1999). Application of positive matrix factorization in source apportionment of particulate pollutants in Hong Kong. Atmospheric Environment 33:3201-3212.

Murphy, D. M., Cziczo, D. J., Froyd, K. D., Hudson, P. K., Matthew, B. M., Middlebrook, A. M., Peltier, R. E., Sullivan, A., Thomson, D. S., and Weber, R. J.: Single-particle mass spectrometry of tropospheric aerosol particles, Journal of Geophysical Research-Atmospheres, 111, D23s32, 10.1029/2006jd007340, 2006.

Martinez, R. E., Williams, B. J., Zhang, Y., Hagan, D., Walker, M., Kreisberg, N. M., Hering, S. V., Hohaus, T., Jayne, J. T., Worsnop, D. R. (2016). Development of a volatility and polarity separator (VAPS) for volatility- and polarity-resolved organic aerosol measurement. Aerosol Science and Technology 50:255-271.

Ng, N. L., Herndon, S. C., Trimborn, A., Canagaratna, M. R., Croteau, P. L., Onasch, T. B., Sueper, D., Worsnop, D. R., Zhang, Q., Sun, Y. L., and Jayne, J. T.: An Aerosol Chemical Speciation Monitor (ACSM) for Routine Monitoring of the Composition and Mass Concentrations of Ambient Aerosol, Aerosol Science and Technology, 45, 780-794, 10.1080/02786826.2011.560211, 2011.

Paatero, P.: Least squares formulation of robust non-negative factor analysis, Chemometrics and Intelligent Laboratory Systems, 37, 23-35, 10.1016/s0169-7439(96)00044-5, 1997.

Ramadan, Z., Song, X. H., Hopke, P. K. (2000). Identification of sources of Phoenix aerosol by positive matrix factorization. J. Air Waste Manage. Assoc. 50:1308-1320.

Sun, Y. L., Zhuang, G. S., Tang, A. H., Wang, Y., and An, Z. S.: Chemical characteristics of PM2.5 and PM10 in haze-fog episodes in Beijing, Environ. Sci. Technol., 40, 3148-3155, 10.1021/es051533g, 2006.

Schauer, J. J., Rogge, W. F., Hildemann, L. M., Mazurek, M. A., Cass, G. R., Simoneit, B. R. T. (1996). Source apportionment of airborne particulate matter using organic compounds as tracers. Atmos. Environ. 30:3837-3855.

[revised manuscript text omitted]

F2

[Figure]

(c)

benzoic acid
furan, 2-ethyl-5-methyl-
nonanoic acid
phthalic acid
triacetin
decanoic acid
3-methylphthalic acid
benzene, p-diacetyl-
phthalimide
dodecanoic acid
2(3H)-furanone,dihydro-5-heptyl-
1-penten-3-one, 1-phenyl-
1,4-dioxaspiro[5,5]undecan-3-one

2(3H)-furanone,dihydro-5-octyl-
heptadecane
benzyl benzoate
phenanthrene
2-propanol, 1-chloro-, phosphate (3:1)
bis(1-chloro-2-propyl)(3-chloro-1-propyl)phosphate
2-Pentadecanone, 6,10,14-trimethyl-
diisobutyl phthalate
nonadecane
2(3H)-furanone,dihydro-5-decyl-
dibutyl phthalate
eicosane
heneicosane
docosane

F3

[Figure]

(d)

benzoic acid
furan, 2-ethyl-5-methyl-
nonanoic acid
phthalic acid
triacetin
decanoic acid
ethanone, 1-[4-(1-methylethenyl)phenyl]-
3-methylphthalic acid
4-methylphthalic acid
undecanoic acid
phthalimide
dodecanoic acid
diethyltoluamide
2(3H)-furanone,dihydro-5-heptyl-
1-penten-3-one, 1-phenyl-
1,4-dioxaspiro[5,5]undecan-3-one

2(3H)-furanone,dihydro-5-octyl-
benzyl benzoate
2-propanol, 1-chloro-, phosphate (3:1)
tris(3-chloropropyl)phosphate
2-Pentadecanone, 6,10,14-trimethyl-
diisobutyl phthalate
nonadecane
2(3H)-furanone,dihydro-5-decyl-
hexadecanoic acid, methyl ester
dibutyl phthalate
eicosane
2(3H)-furanone,dihydro-5-undecyl-
isopropyl palmitate
heneicosane
docosane
tricosane

F4

[Figure]

[Figure]

**Figure 5: The chromatogram profiles of TAG-Bin six-factor solution. The bin width of five scan points and retention time shift correction are used here. The mass spectra in panels d (factor 4) and f (factor 6) are the summed mass spectra of what are largely UCM components in the indicated retention time range.**

[Figure]

**Figure 6: The mass spectra comparison of the six pairs of TAG-Bin (assigned by factor number) vs. AMS – F4 vs. MV-OOA, F6 vs. HOA, F5 vs. SV-OOA, F4 vs. LV-OOA, F1 vs. LOA-AC, F5 vs. LOA2. Each pair shows the maximum Pearson R in Table 2.**
**The mass spectra (m/z 29 – 343 is shown in the color scale) is the normalized signal in log scale.**

[Figure]

Figure 7: The mass spectra of AMS six components.

[Figure]

**Figure 8: Profiles of TAG-Integrated six-factor solution.**

[Figure]

**Figure 9: Profiles of TAG-Integrated six-factor solution.**